# Beating a Dead Horse: On the Redundancies of Adversarial Evaluation of Classifiers

## Abstract

Creating secure systems is challenging; Defenders have to be right all of the time, but attackers only need to be right once. Thus, security evaluations need to employ a variety of attack strategies to identify gaps in the system's defensive posture. In Machine Learning (ML), we often focus our security evaluations on the model, evaluating as many known attacks as possible or using an assumed representative ensemble of attacks to ensure coverage across many possible attack scenarios. However, it is not uncommon for evaluators, *e.g.,* reviewers of a defense proposal, to be presented with a security evaluation resulting from an attack ensemble and still request additional attack evaluations.

In this paper, we study the effectiveness of additional evaluations and re-examine the efficiency of current adversarial robustness evaluation approaches for classification models. Although security evaluations have become increasingly costly due to the increased model scale and dataset size, defensive evaluations still involve running numerous attacks. Even when reviewing an evaluation, additional evaluations may be requested. There is safety in numbers, and what if additional attacks reveal a lack of diversity in the attack scenarios explored by the original evaluation? We examine the question of: "How much more information is learned about the robustness of a defense after the first attack evaluation?". Through three possible lenses of attack diversity, we show that both gradient-based and gradient-free attacks lack any notable variation within their respective classes. A single well-performing attack from each attack class is enough to make a general determination of robustness. When compared to a state-of-the-art and widely used four-attack ensemble, AutoAttack, the simple two-attack ensemble, consisting of one high-performing attack of each class, only differs in evaluation precision by 0.79%.

## 1 Introduction

In today's rapidly evolving technological landscape, Machine Learning (ML) stands at the forefront, driving innovations from healthcare to cybersecurity. As the reliance on ML-based systems grows, evaluating their robustness and security becomes increasingly critical, as vulnerabilities can have far-reaching consequences. One of the most studied ML vulnerabilities, known as evasion attacks (Biggio et al., 2013; Goodfellow et al., 2015; Szegedy et al., 2014), seeks to manipulate an ML model's output by introducing imperceptible manipulations in its inputs.

Creating secure ML models is challenging, as the defender must protect against a wide range of potential attacks, while the attacker only needs to find a single successful attack. This asymmetry creates a significant burden on defenders, who must anticipate and mitigate numerous attack vectors, often with limited resources. Conventionally, the adversarial ML community has relied on attack ensembles to determine the robustness of ML models. Early on, such ensembles contained as many attacks as possible, largely based on their popularity, and defenses that were robust against these ensembles were deemed robust against all attacks within the threat model. However, such ensembles were later shown to be unreliable as the corresponding defenses were compromised by ignored existing attacks, sometimes with small modifications (Engstrom et al., 2018; Athalye et al., 2018; Tramer et al., 2020). To remedy this issue, prior works (Croce & Hein, 2020b; Sheatsley et al., 2023) have proposed ensembles of *diverse* attacks reflecting a wide range of attack strategies.

For example, AutoAttack (Croce & Hein, 2020b) selects attacks that vary their loss function, attack objective formulation, or optimization algorithm. PEA (Sheatsley et al., 2023), instead, exhaustively explore the entire attack space by varying the key components of adversarial attacks observed in the literature, which is more precise than AutoAttack, but also an order of magnitude more costly. Both of these strategies, however, miss the evaluation goal: Is my system secure? We argue that, in most cases, the goal of an adversarial evaluation is to answer this simple binary question. If one evaluation measures the attack success rate to be 41% and another measures it to be 43%, the same conclusion is reached: my system is not secure and can be improved. AutoAttack was novel because its selection of attacks would sometimes reveal large gaps between the reported robustness and the measure robustness, due to gradient obfuscation.

In this paper, we analyze redundancy in adversarial evaluations by examining the diversity among evasion attacks. Rather than characterizing diversity with respect to the attack components, we study it with respect to the attack outcomes:

- *Attack Performance.* Given two or more attacks, what is the improvement in attack success rate as the ensemble increases in size?

- *Inter-class Misclassification.* Given the ground-truth label of an input sample, do different adversarial attacks result in different misclassifications?

- *Adversarial Input Similarity.* Given two attacks or more, how similar are the adversarial inputs they generate?

If two or more attacks with different parameters are similar across all three outcomes, then is there any reason to add them to an ensemble? We leverage the attack decomposition framework of Sheatsley et al. (2023) to examine the two most popular classes of attacks in literature, *i.e.,* gradient-based (Biggio et al., 2013; Goodfellow et al., 2015; Szegedy et al., 2014; Papernot et al., 2016b; Moosavi-Dezfooli et al., 2016; Carlini & Wagner, 2017; Madry et al., 2018; Croce & Hein, 2020a) and gradient-free (Papernot et al., 2017; Chen et al., 2017; Ilyas et al., 2018; Andriushchenko et al., 2020) attacks. Since this framework only supports gradient-based attacks by default, we extend it to support gradient-free attacks by integrating random search optimization. Within both attack classes, we observe a notable lack of diversity across all three attack outcomes. Only when gradient-based attacks are compared to gradient-free ones do we observe a difference in performance outcome. Therefore, in most cases, evaluating with a simple two-attack ensemble containing one attack from each class is sufficient to gain a general understanding of robustness. When compared to AutoAttack (Croce & Hein, 2020b), the most commonly used attack ensemble consisting of four attacks, the evaluation precision of our approach is only 0.79% lower across all unique defenses in the top-10 of the RobustBench leaderboard. Across all the defenses we examine, which includes one gradient obfuscated defense and one defense for the ImageNet dataset, this gap is only 0.75%.

Our **contributions** can be summarized as follows:

- We study the diversity of adversarial attacks and attack ensembles with respect to attack performance, misclassification outcomes, and input perturbations. Across all three outcomes, gradient-based and gradient-free attacks lack diversity within their respective attack classes.

- Through rigorous experimentation with 2 defenses with fundamentally different robustness properties, we show that a single attack from each class is sufficient to obtain a reliable measurement of the defense's robustness. Additional attacks only update the robustness measurement insignificantly.

- We create a two-attack ensemble consisting of a single well-performing attack from each class and compare its robustness measurement to AutoAttack across 10 distinct defenses. Although AutoAttack has twice as many attacks, it reports only a 0.75% higher success rate on average, which does not significantly change one's understanding of a defense's robustness.

- We make our code publicly available at `https://anonymous.4open.science/r/diverse-adv-eval-A2AE` to facilitate efficient adversarial evaluations.

## 2    Background

Most Adversarial ML research focuses on the robustness of image classifiers. An image classifier $f$ maps an image $x \in \mathbb{R}^m$ to a discrete set of labels $\mathcal{Y}$. The classifier learns this mapping using pairs of images and corresponding ground truth labels sampled from a given data distribution $\mathcal{D}$.

### 2.1    Evasion Attacks

Evasion attacks seek to discover imperceptible perturbations $\delta$ that, when added to the input $x$ of the classifier $f$, will cause $f$ to misclassify $x$. Formally, this objective can be defined as follows:

$$\begin{aligned} \text{maximize} \quad & \mathcal{L}_{CE}(f(x+\delta), y_{true}) \\ \text{such that} \quad & \|\delta\|_p \leq \epsilon \\ & x + \delta \in [0,1]^m \end{aligned} \tag{1}$$

Maximizing the classification loss is a reliable way to trigger misclassification. The Projected Gradient Descent (PGD) attack (Madry et al., 2018) exemplifies such *max loss* attacks as it seeks to iteratively solve Equation 1 using the target model's gradients and the Stochastic Gradient Descent optimizer. To avoid getting stuck in local maxima, the perturbation vector is randomly initialized using a standard noise distribution (*e.g.,* uniform noise). Alternatively, some attacks define the adversarial objective based on the norm of the perturbation vector as follows:

$$\begin{aligned} \text{minimize} \quad & \|\delta\|_p \\ \text{such that} \quad & f(x+\delta) \neq f(x) \\ & x + \delta \in [0,1]^m \end{aligned} \tag{2}$$

The Carlini Wagner Attack (CWA) (Carlini & Wagner, 2017) exemplifies such *min norm* attacks as it uses the target model's gradients to iteratively solve Equation 2 using the Adam optimizer. In both cases, the previously mentioned imperceptibility condition is conventionally enforced by constraining the magnitude of the perturbation vector measured using a $\ell_p$-norm function.

In many instances, the target model's gradient may not be usable for attacks. For example, if the attacker does not have internal access to the target model or if the gradients are ill-defined (Athalye et al., 2018). For such cases, several attacks have been proposed that leverage gradient-free optimization algorithms. Notably, the Boundary Attack (Brendel et al., 2018) and the Square attack (Andriushchenko et al., 2020) develop random search optimization algorithms to identify adversarial perturbations. Such algorithms iteratively sample perturbations from a carefully crafted distribution, retain it if the attack objective improves, and discard it otherwise. Prominent attacks that follow this strategy primarily differ in how they design the sampling distribution. Besides random search, other gradient-free strategies involve approximating target model's gradients to perform a gradient-based attack, using methods like finite differences (Chen et al., 2017), natural evolution strategies (Ilyas et al., 2018), and transferring adversarial samples from surrogate models (Papernot et al., 2016a; Liu et al., 2016; Papernot et al., 2017; Tramèr et al., 2017).

### 2.2    Defenses Against Evasion Attacks

Despite widespread awareness of the threat of evasion attacks on AI systems, there appears to be a surprising lack of effective and practical methods to fortify systems against these attacks. One of the most popular strategies to defend against evasion attacks is to simply "train with more data" (Madry et al., 2018; Zhang et al., 2019). Adversarial training (Madry et al., 2018) augments the training data with adversarially perturbed examples. By exposing the model to crafted adversarial inputs during training, adversarial training aims to improve the model's ability to correctly classify perturbed instances and thereby fortify its resilience against potential evasion attacks. Alternate strategies involve modifying the loss function of the classifier to enforce robustness (Wan et al., 2018; Pang et al., 2019a), altering the classification pipeline to destroy the information carried by the adversarial noise as the perturbed input propagates through it (Xiao et al., 2020;

**Table 1:** Attack implementations available in popular Adversarial ML toolkits. Gradient-based attacks dominate across all toolkits.

| Toolkit | Evasion | | Poisoning | Inference | Extraction |
| | w/ Grads | w/o Grads | | | |
| --- | --- | --- | --- | --- | --- |
| ART (Trusted-AI, 2023) | 28 | 12 | 11 | 6 | 3 |
| CleverHans (CleverHans-Lab, 2021) | 14 | 4 | 0 | 4 | 3 |
| Foolbox (Bethge-Lab, 2024) | 16 | 2 | 0 | 0 | 0 |
| advertorch (Borealis-AI, 2022) | 12 | 2 | 0 | 0 | 0 |
| Counterfit (Azure, 2022) | 10 | 3 | 0 | 0 | 0 |

Verma & Swami, 2019; Pang et al., 2019b; Sen et al., 2019), or employing generative modeling (Samangouei et al., 2018; Song et al., 2018; Li et al., 2019).

### 2.3 Evaluating Defenses Against Evasion Attacks

Assessing the vulnerability of a defense in an absolute manner requires evaluating it against all known attacks that exploit the specific vulnerability. Unfortunately, conducting such a comprehensive assessment for evasion vulnerabilities is computationally prohibitive. This is because evasion is the most extensively studied vulnerability, as evidenced by the data in Table 1, which reports the number of attack implementations available in popular toolkits for various types of vulnerabilities. The second best option is to evaluate against an ensemble of diverse attacks. In the early days of Adversarial ML research, there was a lack of agreement on which attacks would comprise such an ensemble. As a result, several defense evaluators fell into the trap of evaluating against an ensemble of very similar attacks (*e.g.,*, FGSM (Szegedy et al., 2014), BIM (Kurakin et al., 2018), and PGD (Madry et al., 2018)) and claiming robustness broadly against all attacks. This lack of "diversity" among attacks led to these evaluations being deemed unreliable by follow up works (Engstrom et al., 2018; Athalye et al., 2018; Tramer et al., 2020; Croce & Hein, 2020b), and the corresponding defenses being deemed non-robust. As the field progressed, new ensembles were developed and trusted for reliable adversarial evaluations.

***Ensemble of Diverse Attacks.*** The practice of using diverse attacks for reliable evaluations first gained traction with the use of the combination of Projected Gradient Descent (Madry et al., 2018) attack and the Carlini-Wagner attack (Carlini & Wagner, 2017). Here, diversity is defined based on the formulation of the adversary's objective, *i.e., max loss* or *min norm*. Currently, the most popularly used ensemble of attacks is the AutoAttack (Croce & Hein, 2020b). Retaining the previously mentioned definition of diversity, AutoAttack incorporates state-of-the-art *max loss* (APGD-CE (Croce & Hein, 2020b)) and *min norm* (FAB (Croce & Hein, 2020a)) attacks. Furthermore, AutoAttack expands upon this definition by adding distinct loss functions (APGD-DLR (Croce & Hein, 2020b)) and optimization algorithms (Square Attack (Andriushchenko et al., 2020)) to the ensemble.

More recently, Sheatsley et al. (2023) propose an attack decomposition framework for gradient-based attacks. Leveraging their framework, they obtain a set of 432 unique attacks.[1] Furthermore, they propose an attack called the Pareto Ensemble Attack (PEA) that exhaustively explores this large set of attacks to obtain a precise measure of robustness against said set. However, such an exhaustive search is computationally expensive.

## 3 Diversity-driven Adversarial Evaluations

*What is the goal of an adversarial evaluation?* There are several possible answers to this question:

---

[1]While the original paper used 576 attacks, the released code includes a modified framework with 5 components, yielding a total of 432 attacks.

- Making a *precise* measurement of robustness in such a way that it represents the true worst-case performance of the model as faithfully as possible;

- Making a *general* determination of robustness within an acceptable error margin;

- Measuring robustness only within *context-dependent* high risk situations.

When it comes to machine learning security, how useful is it to obtain a precise measurement compared to the other two goals? Consider a road sign classification model:

- A general robustness evaluation gives the same understanding as a precise robustness evaluation. Knowing that a road sign classification model has an 80% error rate versus an 81.23% error rate when under attack is the same: the model is unreliable.

- A context-dependent robustness evaluation provides a more nuanced answer. Knowing that a road sign classification model has a 1% error rate on recognizing stop signs versus an 81.23% overall error rate when under attack informs the user that in a specific high-risk scenario, the model is unreliable.

While many may share this understanding, precise robustness evaluations are often requested. The community is rightfully skittish when reviewing evaluations of adversarial defenses after the failure of many early gradient obfuscation works. To prevent bad evaluation, the most often request is for defenders to design "adaptive evaluations", which are evaluations customized to the specific defense. However, a well-designed adaptive evaluation can be time-consuming, and defenders are not incentivized to break their own defense as the community often does not accept a presentation on experimental failures and the lessons learned. For most defenses developing an adaptive attack may require non-trivial effort, that is outside the scope of their work. The alternative is to empirically evaluate a defense by exploring various attack scenarios, *i.e.,* varied attack components, within the threat model. If the attack ensemble is diverse enough, then the evaluation can be trusted. Unfortunately, the meaning of attack diversity is unclear. Even for proposed defenses that followed community evaluation guidelines, they were rejected for not exploring more of the attack space and convincing reviewers of sufficient attack diversity (Dhaliwal & Hambrook, 2019; Chen et al., 2019). This leads to the question of: what does it mean for an attack to be diverse?

Defining attack diversity based on attack components (loss function, optimization algorithm, random start strategy, hyper-parameters, etc.) is imprecise, as two attacks with different parameters could result in the same robustness measurement, even when combined. As the goal of an evaluation is to learn about the defense, we propose defining diversity with respect to attack outcomes rather than attack components. There are three factors in which attack outcomes can differ:

- *Attack Success Rate.* Different attack techniques may specialize in attacking different parts of the input space or bypassing certain defensive techniques. For example, gradient-free attack methods are useful for bypassing gradient-shattering defenses.

- *Inter-class Misclassification.* Different attacks may result in the same adversarial accuracy, but cause different misclassifications for the same input.

- *Adversarial Input Similarity.* Different attacks may result in the same adversarial accuracy but generate dissimilar adversarial inputs in the input space. For example, image classifiers can be compromised with imperceptible perturbations applied to the entire image as well as perceptible perturbations applied to a localized patch, both of which need to be prevented.

With these definitions in mind, we will study existing gradient-based and gradient-free attacks and analyze the diversity of ensembles created from these attacks. When chosen properly, does increasing the number of attacks in an ensemble improve one's understanding of robustness based on the possible attack outcomes?

### 3.1 Experimental Setup

To facilitate our analysis, we use the attack decomposition framework of Sheatsley et al. (2023) to obtain a large set of gradient-based attacks. To obtain a set of gradient-free attacks, we expand their framework to enable support for a gradient-free optimization algorithm. Specifically, we integrate the random search algorithm of Square Attack (Andriushchenko et al., 2020), which allows us to cover all of the attacks in the AutoAttack (Croce & Hein, 2020b) ensemble using their framework.

***Attack Details.*** With minor improvements to the framework from Sheatsley et al. (2023), we obtain a set of 144 unique $\ell_\infty$ gradient-based attacks and 20 unique $\ell_\infty$ gradient-free attacks. Each gradient-based attack is run for 100 steps with 5 random restarts, and each gradient-free attack is run for 2000 steps (or queries) with no random restarts. These hyperparameters are borrowed from AutoAttack (Croce & Hein, 2020b), who deemed them sufficient for the proper convergence of the attack's objective.

***Evaluated Defenses.*** In Sections 3.2 to 3.3, we analyze the diversity in our sets of gradient-based and gradient-free attacks based on the three different definitions previously described. A single comprehensive evaluation requires running all 164 unique attacks to completion. Limited evaluation resources necessitate that we carefully select which defenses to comprehensively evaluate to ensure diversity among defensive scenarios and better inform our later experiments. In general, there are two defensive scenarios: the gradient is present and well defined or it is not. We choose to evaluate a ResNet-18 classifier trained with the FastAT defense (Wong et al., 2019), which has well-defined gradients, and a ResNet-18 classifier protected by the k-winners-take-all defense (Xiao et al., 2020), which obfuscates the gradient. Most other defenses may vary in their technique but can be classified into one of these scenarios. Both defenses were trained to be robust against an $\ell_\infty$ adversary with the budget of $\epsilon = 8/255$. Note that we rely on the pre-trained model checkpoints rather than training the defenses ourselves.

***Evaluation Metrics.*** Throughout our experiments, we measure the performance of an attack in terms of success rate, *i.e.,* the percentage of test samples on which the attack succeeds. Following Sheatsley et al. (2023), we treat the success rate obtained using PEA as the highest success rate attainable using a given set of attacks.

### 3.2 Attack Success Rate Diversity

A diverse attack ensemble can be created by combining complementary attacks that work together to bypass various defensive techniques. We begin by exploring the construction of such ensembles, using cumulative success rate as the guiding metric. At first glance, it may not be obvious which attack, from the broad range available in the literature, will be most effective against a given defense. This uncertainty, coupled with the pressure to meet reviewers' often vague expectations for adaptive attack evaluations (Dhaliwal & Hambrook, 2019; Chen et al., 2019), leads many defense authors to evaluate their methods using as many attacks as possible. Since the majority of available attacks are gradient-based (as shown in Table 1), most evaluations heavily rely on these types of attacks. However, the value of employing multiple attacks from the same "class" in determining a defense's true robustness remains unclear. Thus, we first investigate whether empirical evidence supports the inclusion of multiple attacks from the same class in diverse ensembles.

First, we obtain the individual success rate of every possible gradient-based and gradient-free attack. Then, starting with a single attack, we greedily add one attack at a time to our ensemble using the ensemble's cumulative success rate as the criteria. In Figure 1, we plot the success rate of the ensemble of gradient-based attacks *vs.* the number of attacks in it. The overall success rate of PEA, *i.e.,* the highest attainable success rate using a given set of attacks, is plotted as the green dashed line. We observe that $\sim 99\%$ of PEA's success rate can be attributed to a single attack. Furthermore, an ensemble of 6 attacks out of 144 possible attacks is sufficient to achieve PEA's success rate. We observe similar results for gradient-free attacks as shown in Figure 1b. A majority of PEA's success rate, *i.e.,* $\sim 93\%$, can be attributed to a single attack. Collectively, our results suggest that there is little meaningful diversity among gradient-based and gradient-free attacks with respect to attack performance. Unless a precise robustness measurement is required, a single strong attack in each class is sufficient to understand the strength of a proposed defense.

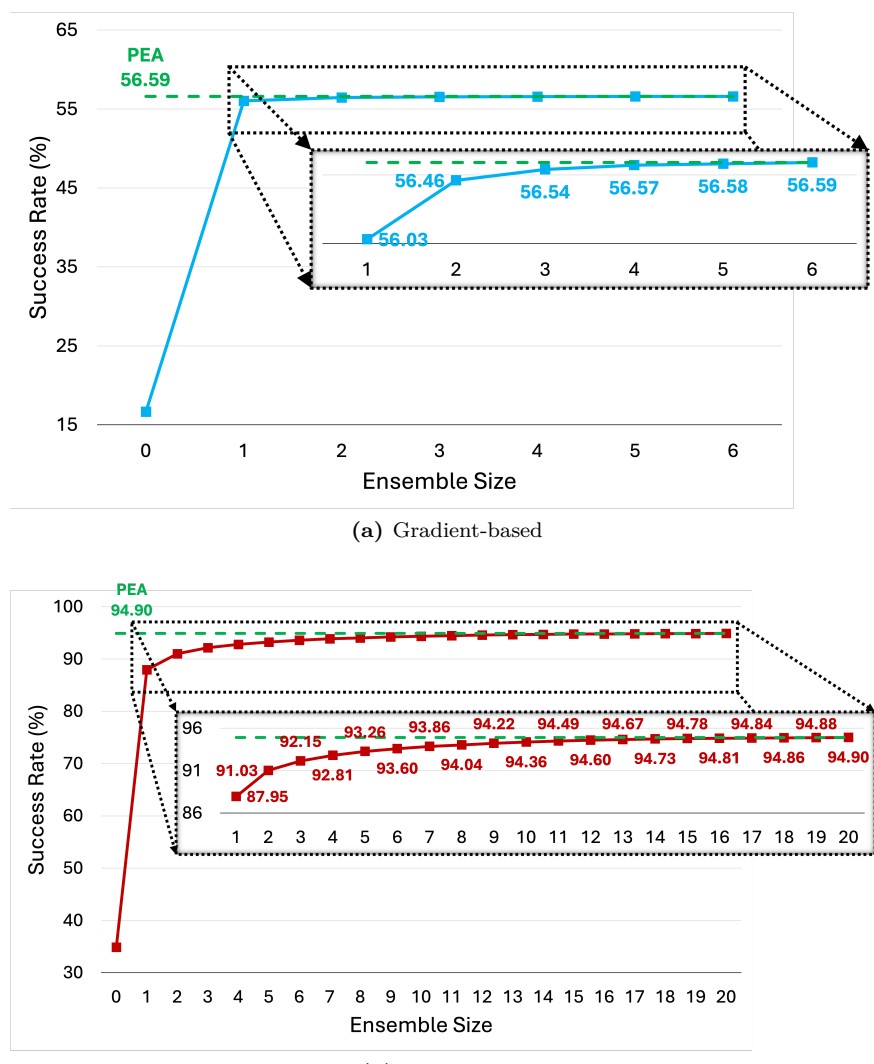

**(a)** Gradient-based

**(b)** Gradient-free

**Figure 1:** Plotting the cumulative success rate of an ensemble versus the number of attacks for (a) gradient-based attacks and (b) gradient-free attacks reveals an important trend. Increasing the diversity of an ensemble by adding new attacks from the same class results in diminishing gains in cumulative success rate.

***Evaluating with Individual Attacks.*** Although a single attack contributes greater than 90% of the ensemble's success rate, *are all attacks created equal?* Yes and No. We plot the percentage of attacks that achieve greater than a specified percentage of the maximum attainable success rate in Figure 2. There is a notable variation in success rates across all attacks. A possible explanation for this variation is incompatibility between components that form certain attacks. However, there are groups of attacks that perform similarly well. Approximately 47% of gradient-based attacks achieve over 90% of the success rate of PEA. For gradient-free attacks, 65% of attacks reach this success rate. Overall, we find that within each class of attacks, there exist several individual attacks that can be relied upon to make a general determination of a defense's robustness.

***Comparison with AutoAttack.*** We further investigate the reliability of individual attacks when the goal is to make a general determination of a defense's robustness. Depending upon the quality of the target classifier's gradients, we compare the success rate of either a single gradient-based (GB-ATK)[2] or a

---

[2]This attack uses the following components: (i) BackwardsSGD optimizer; (ii) MaxStart initialization; (iii) Cross-entropy loss; and (iv) DeepFool saliency map. For further details regarding each component, please refer to the work by Sheatsley et al. (2023)

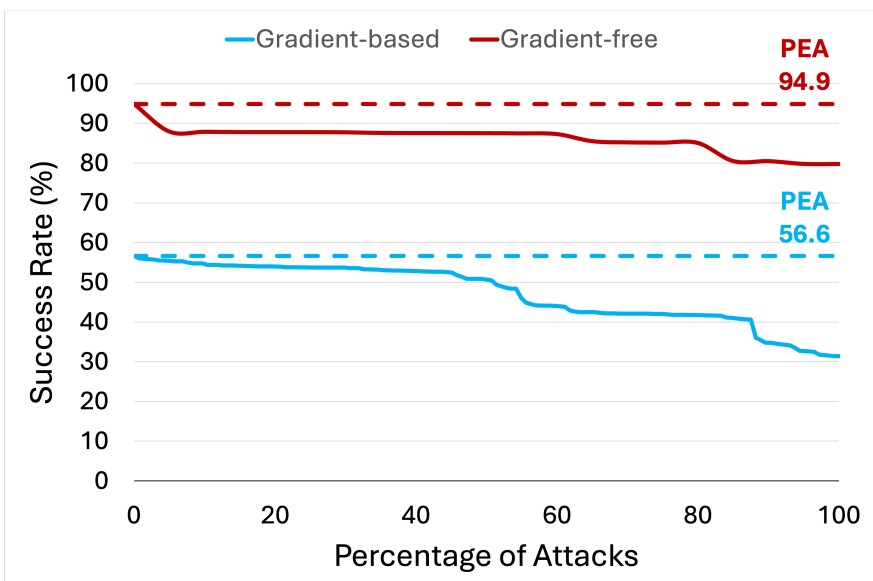

**Figure 2:** Plotting the percentage of **gradient-based** and **gradient-free** attacks achieving a success rate greater than a given value. We find that a large proportion of attacks can be relied upon as representatives of their respective classes.

**Table 2:** Investigating the reliability of individual gradient-based and gradient-free attacks (GB-ATK and GF-ATK) in accurately determining the robustness of defenses. We compare the success rate of these attacks against AutoAttack across multiple distinct defenses. Individual attacks remain competitive with AutoAttack ensemble irrespective of defense methods, model architectures, and datasets used.

| Defense | Attack | Success Rate (%) |
|---|---|---|
| Wong2020Fast (Wong et al., 2019) | AutoAttack | 56.50 |
| | GB-ATK | 55.22 |
| Peng2023Robust (Peng et al., 2023) | AutoAttack | 28.82 |
| | GB-ATK | 28.54 |
| Salman2020Do_R18 (Salman et al., 2020) | AutoAttack | 69.18 |
| | GB-ATK | 69.38 |
| Xiao2020Enhancing (Xiao et al., 2020) | AutoAttack | 88.51 |
| | GF-ATK | 87.76 |

single gradient-free attack (GF-ATK)[3] against that of AutoAttack. In addition to the two defenses used so far (Wong et al., 2019; Xiao et al., 2020), we report results for the following two defenses: (i) one defense from the top-10 of the CIFAR-10 leaderboard of RobustBench (Peng et al., 2023) and (ii) a defense (Salman et al., 2020) for the ImageNet dataset. For both defenses, we use pre-trained weights available in the RobustBench (Croce et al., 2021) model zoo. For the CIFAR-10 defenses, we use $\epsilon = 8/255$, and for the ImageNet defense, we use $\epsilon = 4/255$. When evaluating with ImageNet, we only use 5000 samples from the validation set for evaluation following Croce et al. (2021). The corresponding results are reported in Table 2. Against all defenses, a single attack is able to achieve more than 99% of the success rate of AutoAttack. Although the community relies on AutoAttack to make a general determination of a defense's robustness, only a single attack is sufficient as the extra few percent higher success rate of AutoAttack doesn't change our determination.

---

[3]This attack uses the following components: (i) RandomSearch optimizer; (ii) RandomPatch initialization; and (iii) Margin loss.

> **Takeaway I**: A single attack from each of the gradient-based and gradient-free classes is sufficient to obtain a general determination of a defense's robustness against that class.

### 3.3 Inter-class Misclassification Diversity

If top-performing attacks do not widely different in attack success rate, maybe they will vary in the adversarial predictions they generate. Creating an ensemble based on this definition of diversity can help a defender identify potential weaknesses and implement appropriate countermeasures in advance. Therefore, we analyze diversity within the two classes of attacks from the previous section on the basis of the misclassifications they cause, *i.e.,* diversity in the output space of the target model. Given a test sample, we measure how often successful attacks produce the same adversarial prediction. There are 55.6% and 37.9% of samples that had

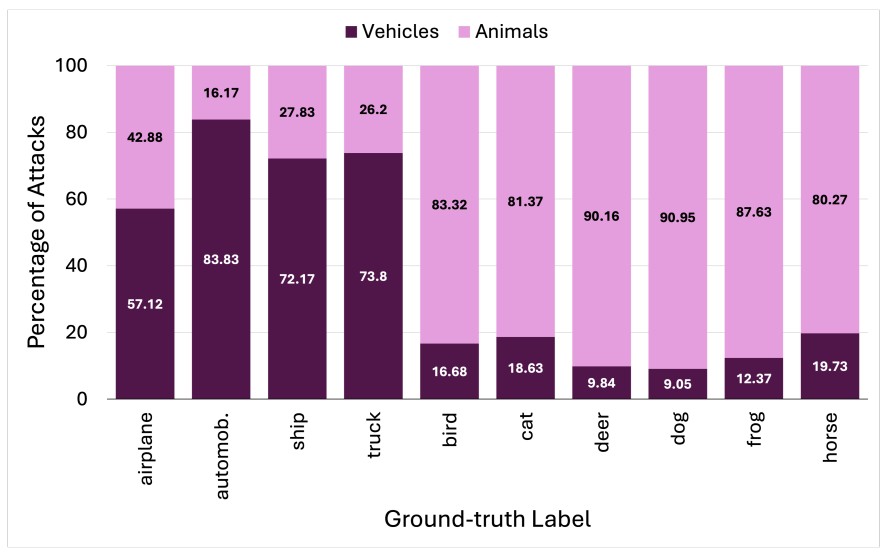

**(a)** Gradient-based

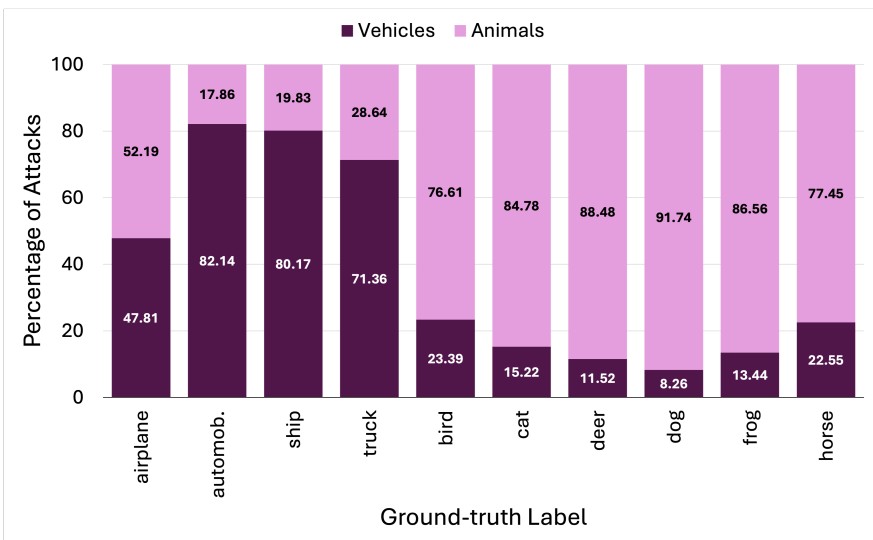

**(b)** Gradient-free

**Figure 3:** Plotting the distribution of adversarial predictions per ground-truth label over the two broad semantic categories in the CIFAR-10 dataset, *i.e.,* animals and vehicles. Attacks tend to generate predictions within classes that are semantically similar to the ground-truth class. This observation remains consistent for both (a) gradient-based attacks and (b) gradient-free attacks.

**Table 3:** The top-2 adversarial predictions per ground-truth label, along with the percentage of attacks that result in these predictions. Top-2 predictions remain in the same semantic category as the ground-truth label for both gradient-based and gradient-free attacks.

| Ground-truth | | Top-2 Predictions | | | |
|---|---|---|---|---|---|
| | | Gradient-based | | Gradient-free | |
| **Vehicles** | airplane | ship: 40.7 | bird: 14.8 | ship: 34.94 | bird: 29.55 |
| | automobile | truck: 61.0 | ship: 15.1 | truck: 53.18 | ship: 20.08 |
| | ship | airplane: 40.3 | truck: 19.5 | airplane: 50.97 | truck: 15.36 |
| | truck | automobile: 45.7 | ship: 17.2 | automobile: 43.48 | ship: 16.83 |
| **Animals** | bird | deer: 26.7 | frog: 21.9 | deer: 23.55 | cat: 17.09 |
| | cat | dog: 34.7 | frog: 21.2 | dog: 35.72 | deer: 13.34 |
| | deer | frog: 28.8 | bird: 21.0 | bird: 29.95 | horse: 21.87 |
| | dog | cat: 50.7 | frog: 13.6 | cat: 51.09 | horse: 16.15 |
| | frog | deer: 34.9 | cat: 24.8 | deer: 29.65 | cat: 23.36 |
| | horse | deer: 30.7 | dog: 17.6 | deer: 29.92 | dog: 19.41 |

the same adversarial prediction from all successful attacks within gradient-based and gradient-free attacks, respectively.

Since we are using the CIFAR-10 dataset, we can divide all classes into two semantic categories: vehicles and animals. We hypothesize that not only are the adversarial predictions similar per sample, but they are also similar per semantic category. We plot the percentage of attacks that return adversarial predictions within the same semantic category as the ground-truth label (in Figure 3). We find that untargeted attacks tend to remain within the same semantic category as the ground-truth label. In the case of gradient-based attacks, across all images of animals, 85.6% of attacks return adversarial prediction that remains within the animal category. For images of vehicles, 71.7% of attacks return adversarial predictions that remain within the vehicle category. In the case of gradient-free attacks, these numbers are 84.3% and 70.4%, respectively.

Interestingly, airplanes get misclassified as animals slightly more often than vehicles in case of gradient-free attacks. To understand why, we look at the top-2 most frequent adversarial predictions per ground-truth label in Table 3. Airplanes tend to get misclassified as birds, potentially because a sky background occurs commonly across both these categories. Collectively, these observations suggest that there isn't meaningful diversity within both classes of untargeted attacks in terms of the adversarial predictions they generate.

> **Takeaway II**: Untargeted gradient-based and gradient-free attacks tend to cause the same, easy-to-achieve misclassifications.

### 3.4 Adversarial Input Diversity

We examine our final diversity category, diversity with respect to the adversarial inputs each attack generates. Although adversarial inputs might appear visually similar to a human, there could be subtle nuances in the data itself that only the classifier picks up on. For each benign test sample, we compute the pairwise distances between the adversarial samples generated by all unique pairs of successful attacks. We do this for all test samples and plot the histogram of pairwise cosine distances in Figure 4. Given that Euclidean distance is inappropriate for measuring similarity between high-dimensional points due to the curse of dimensionality, we measure similarity using cosine distance instead. Cosine distance ranges from 0 to 2, where 0 indicates identical vectors, 1 indicates perpendicular vectors, and 2 indicates completely opposite vectors.

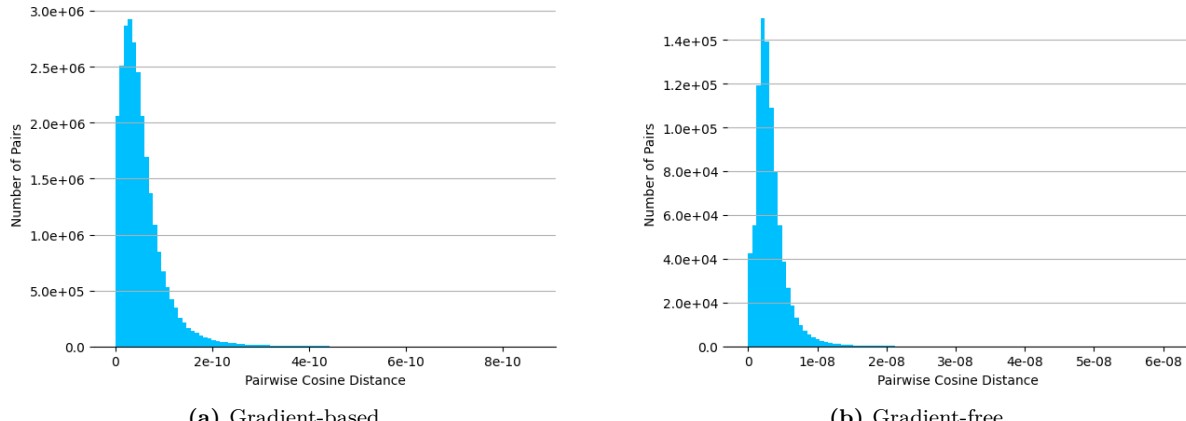

**(a)** Gradient-based                    **(b)** Gradient-free

**Figure 4:** Plotting the histogram of pairwise cosine distances between all unique pairs of adversarial samples generated for a given benign sample. Both (a) gradient-based and (b) gradient-free attacks perturb any given benign input very similarly in terms of cosine distance.

For both gradient-based and gradient-free attacks, we find that the histogram of pairwise distances is concentrated at near-zero values. In the case of gradient-based attacks, the peak occurs at $2.2 \times 10^{-11}$, whereas in the case of gradient-free attacks, the peak occurs at $1.5 \times 10^{-9}$. We observe a similar distribution for pairwise cosine distances between adversarial perturbations. Since the majority of pairwise distances are extremely close to 0, this indicates that the majority of adversarial samples generated for a given benign sample are very similar.

> **Takeaway III**: For a given input, gradient-based and gradient-free attacks tend to generate very similar adversarial perturbations within their respective classes.

## 4    Redundancies in Attack Ensembles

In Section 3, we used three distinct definitions of diversity to demonstrate that gradient-based and gradient-free attacks exhibit limited diversity within their respective classes. Specifically, attacks from the same class tend to (i) succeed on the same subset of inputs, (ii) induce the same misclassification labels for a given input, and (iii) produce highly similar adversarial perturbations in the $\ell_2$ norm space. As a result, running multiple attacks from the same class is often redundant, offering little additional insight into a defense's vulnerabilities. In this section, we investigate redundancy in prominent attack ensembles, namely AutoAttack (Croce & Hein, 2020b) and the Pareto Ensemble Attack (PEA) (Sheatsley et al., 2023). Our analysis reveals inefficiencies in these ensembles, showing that a large portion of their overall success rate can be retained using only a subset of their constituent attacks.

### 4.1    AutoAttack (Croce & Hein, 2020b)

For both gradient-based and gradient-free attacks, we found that a single attack is sufficient for making a general determination of a defense's robustness against that class. But which attack class should we evaluate against? The conventional wisdom that gradient-based attacks are strictly stronger than gradient-free attacks has long been invalidated. Therefore, it may not always be evident *a priori* which attack class a given defense is most vulnerable to, and using the wrong class can lead to unreliable evaluations. For example, in the case of defenses with obfuscated gradients, using more gradient-based attacks was not the solution; switching to a single gradient-free attack was. In Figure 5, we evaluate the k-winner-takes-all defense (Xiao et al., 2020) using gradient-based attacks only. No matter the size of the ensemble, we observe only minor improvements in adding more attacks. Once a single gradient-free attack is added, the success rate of the ensemble is greatly increased.

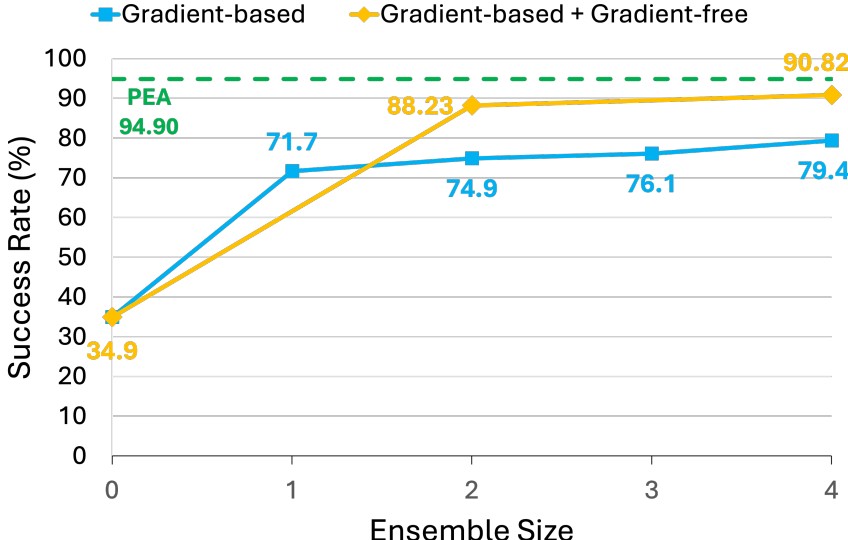

**Figure 5:** Plotting the cumulative success rate of an ensemble *vs.* the number of attacks in it. Increasing the diversity of an ensemble by adding new gradient-based attacks doesn't provide an accurate determination of the robustness of a defense whose gradients are ill-defined. On the other hand, increasing attack diversity by adding a single gradient-free attack is sufficient.

Croce & Hein (2020b) were the first to identify the value of such diversity, using it to motivate the inclusion of Square Attack within the AutoAttack ensemble. However, AutoAttack also has three gradient-based attacks. Our diversity analysis from before suggests that while adding Square Attack makes sense, the three gradient-based attacks are likely redundant if the goal is to make a general determination of a defense's robustness. When referring to Table 2 in the AutoAttack paper, we observe that the difference between the AutoAttack success rate and the gradient-based attack success rate is less than 1% across most defenses, with APGD-DLR often reporting the most accurate result. Therefore, AutoAttack could be more efficient if reduced to two attacks and still provide evaluators with the same general understanding of a defense. We test this by comparing the success rate of AutoAttack with an ensemble with a single well-performing gradient-based and gradient-free attack chosen from our experiments earlier. In addition to the defenses evaluated earlier, we also evaluated all unique defenses from the top-10 of the RobustBench leaderboard for CIFAR-10[4] and reported them in Table 4. Across all defenses, our two ensemble evaluation has only 0.75% lower success rate than AutoAttack on average. This highlights the redundancy in AutoAttack.

### 4.2 PEA (Sheatsley et al., 2023)

In the previous section, we observed that both gradient-based and gradient-free attacks lack meaningful distinction within them according to three goal-oriented definitions of diversity. As a result, a single attack from each class is sufficient to make a general determination of robustness against that class. Collectively, an ensemble of a gradient-based and a gradient-free attack is sufficient to make such a determination, irrespective of defense strategy, classifier architecture, or dataset used.

But what if the goal of the adversarial evaluation is to make a robustness measurement that is as precise as possible? Currently, the community relies on AutoAttack to achieve this goal. However, AutoAttack may not provide the most precise measurement as it does not exhaustively sweep the attack space. PEA (Sheatsley et al., 2023), on the other hand, returns the highest success rate possible from a set of attacks as it exhaustively searches the entire attack set till it finds an attack that succeeds.[5] Therefore, it represents the most precise ensemble available in the literature. While iterating over all attacks makes PEA precise, this also

---

[4]Leaderboard version: `https://github.com/RobustBench/robustbench/tree/78fcc9e48a07a861268f295a777b975f25155964`
[5]In the original paper, PEA is defined over the set of all gradient-based attacks, similar to the one we examined in previous sections.

**Table 4:** Comparing the success rate of an ensemble of two attacks – a gradient-based (GB-ATK) and a gradient-free (GF-ATK), against AutoAttack – an ensemble of three gradient-based and a gradient-free attack. Evaluation is performed on all unique defenses from the top-10 of the Robustbench leaderboard. We also report results for one ImageNet defense and the defenses from the previous sections. The two-attack ensemble is comparable in performance to AutoAttack across all defenses. Therefore, the two additional gradient-based attacks are redundant and only incrementally contribute to the robustness measurement.

| Rank | Defense | Success Rate (%) | | |
|---|---|---|---|---|
| | | AutoAttack | GB-ATK + GF-ATK | Δ |
| 1 | Bartoldson2024Adversarial (Bartoldson et al., 2024) | 26.29 | 26.00 | 0.29 |
| 2 | Amini2024MeanSparse (Amini et al., 2024) | 26.90 | 24.65 | 2.25 |
| 4 | Peng2023Robust (Peng et al., 2023) | 28.93 | 28.54 | 0.39 |
| 5 | Wang2023Better (Wang et al., 2023) | 29.31 | 28.89 | 0.42 |
| 6 | Bai2024MixedNUTS (Bai et al., 2024b) | 30.29 | 28.52 | 1.77 |
| 8 | Bai2023Improving (Bai et al., 2024a) | 31.94 | 31.42 | 0.52 |
| 9 | Cui2023Decoupled (Cui et al., 2024) | 32.27 | 31.99 | 0.28 |
| 74 | Wong2020Fast (Wong et al., 2019) | 56.50 | 55.23 | 1.27 |
| - | Xiao2020Enhancing (Xiao et al., 2020) | 88.51 | 88.08 | 0.43 |
| 29 | Salman2020Do (Salman et al., 2020) (ImageNet) | 69.18 | 69.40 | 0.22 |

makes it computationally intensive. Given that many researchers, developers, and a significant portion of the academic community operate under compute-restricted settings, running PEA may not always be feasible.

PEA scales poorly with the size of the set of attacks it exhaustively searches. This issue is exacerbated by the increase in model and dataset size. Previously, we showed in Figure 2 that some attacks have suboptimal success rates. Therefore, it is likely that such attacks do not make little to no contributions to PEA's final success rate as their results overlap with other, more optimal attacks. In this section, we investigate how redundant the PEA ensemble really is.

### 4.2.1 Analysis

To perform this investigation, we study how effective PEA will be if we drop attacks based on their effectiveness during evaluation. Our hypothesis is that only the top few performing attacks contribute to PEA's eventual success rate, and hence, dropping low-performing attacks will save compute while having a negligible effect on attack success rate. Specifically, we attack one sample at a time using all available attacks and record their success rate. If an attack fails more than $N$ times, we drop it from our set of attacks. We continue this process till all samples from the test set have been processed. The final success rate is the number of samples on which at least one attack succeeded. This analysis methodology is summarized in Algorithm 1. We perform our analysis by using PEA with two different sets of attacks – our gradient-based and gradient-free attacks from the previous sections. We apply the gradient-based PEA on FastAT (Wong et al., 2019) and the gradient-free PEA on k-winners-take-all (Xiao et al., 2020).

We refer to $N$ as the number of lives of an attack, and it acts as a knob to control the effort required to evaluate using the ensemble. We treat the cumulative number of attacks executed during an evaluation as a quantitative proxy for the "effort". Changing the value of $N$ from low to high results in an increase in the cumulative number of attacks executed and, therefore, the overall effort of the evaluation (see Figure 6). Next, we ask, how much of this effort is actually useful towards the final success rate of the ensemble? And how much of it is redundant?

In Table 5, we report the success rate obtained using each value of $N$, along with the percentage of attacks executed relative to PEA. As the value of $N$ increases, we gradually get closer to PEA's success rate. In the case of FastAT, we are able to achieve all of PEA's success rate while running only 84.79% of attacks relative

---

**Algorithm 1** Executing PEA (Sheatsley et al., 2023) while dropping attacks

---

**Require:** Robust ML model
**Require:** List of benign samples correctly classified by it
**Require:** Set of attacks with initial lives $N$
 1: *winner* ← **false**
 2: *losers* ← empty set
 3: *success_rounds* ← 0
 4: Randomly shuffle the set of samples
 5: **for** each sample $s$ in the shuffled set of samples **do**
 6:     Randomly shuffle the set of attacks
 7:     **for** each attack $a$ in the shuffled set of attacks **do**
 8:         **if** attack $a$ succeeds on sample $s$ **then**
 9:             *winner* ← **true**
10:             increase *success_rounds* by 1
11:             **break**
12:         **else**
13:             add $a$ to *losers*
14:         **end if**
15:     **end for**
16:     **if** more than one attack remain AND *winner* == **true then**
17:         **for** each attack $a$ in *losers* **do**
18:             Decrease the lives of attack $a$ by 1
19:             **if** lives of attack $a$ == 0 **then**
20:                 Remove attack $a$ from the set of attacks
21:             **end if**
22:         **end for**
23:     **end if**
24: **end for**
25: **return** *success_rounds* / total number of samples

---

to it. In the case of k-winners-take-all, it takes 91.67% of attacks relative to PEA to achieve its success rate. More attacks are required in this case because, unlike in the case of FastAT, all attacks against k-winner-takes-all make a non-zero contribution to the final success rate (recall Figure 1). In such extreme cases, a precise measurement will require running all attacks. These results clearly highlight the redundant effort required by PEA and validate our hypothesis that the top few performing attacks contribute the majority of the ensemble's success rate.

## 5  Limitations

In this section, we address the limitations of our study in an effort to better specify the scope of our current work, as well as propose interesting future works.

***Targeted Attack.*** We did not study how targeted attacks affect the results of an adversarial evaluation, specifically, when looking to address context-dependent robustness. In certain scenarios, we may be more concerned with a specific type of adversarial misclassification due to the implied risk such as a road sign classifier mistaking a stop sign sign as a speed limit sign, rather than general misclassification. With respect to this goal, we do not have reason to believe that targeted attacks would differ in their lack of diversity compared to one another. Except for the specification of a misclassification label, the attack mechanisms remain the same as with untargeted attacks. We hypothesize that a general determination of context-dependent robustness can be obtained via a single gradient-free and a single gradient-based targeted attack, but further experimentation is needed.

With respect to including a targeted attack, gradient-free or gradient-based, when making a general determination of robustness, our comparisons with AutoAttack suggest that evaluation precision is only slightly

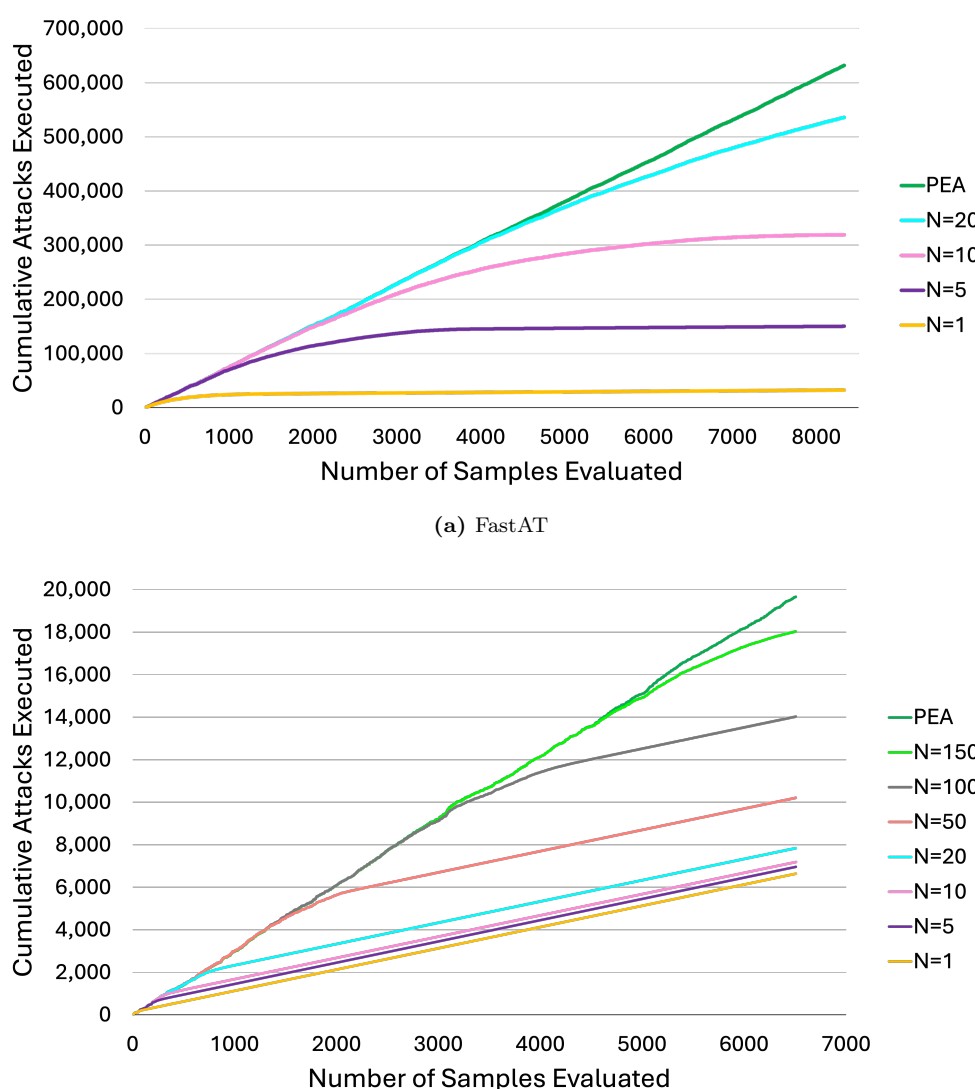

**(a)** FastAT

**(b)** k-winners-take-all

**Figure 6:** Plotting the cumulative number of attacks executed for different values of $N$ (number of attack lives) when evaluating (a) FastAT and (b) k-winners-take-all. Higher values of $N$ leads to higher cumulative attacks executed and, therefore, higher computational cost.

increased. AutoAttack uses a targeted attack to enumerate across the possible attack directions, which occasionally succeeds on some inputs that failed untargeted attacks were not able to perturb.

Targeted attacks can also improve the precision of adversarial evaluations with respect to the first two evaluation goals. For a given sample, the targeted attack enumerates across all of the possible class labels, excluding the original prediction, and selects the first successful attack. This approach is used by AutoAttack (Croce & Hein, 2020b) to improve the precision of the evaluation results. However, as we demonstrated in Tables 2 and 4, this improvement is marginal compared to a two-attack ensemble of a single gradient-based attack and a single gradient-free untargeted attack.

***Adaptive Attacks.*** Adaptive attacks, expertly customized attacks tailored to bypass certain aspects of a defense, are a common request when studying a potential defense. Tramer et al. (2020) detail several such defense-specific adaptations, for example, identifying and targeting the weakest parts of a defense, or adapting the attack objective to improve its convergence. As these attacks are tailored to target a specific defensive component, it is difficult to automate this process and would likely only be useful if the targeted

**Table 5:** Success rate and percentage of attacks required to achieve it when running PEA using different values of $N$ (number of attack lives). We report mean and standard deviation across five independent trials.

| | FastAT (Wong et al., 2019) | | | k-winners-take-all (Xiao et al., 2020) | |
|---|---|---|---|---|---|
| $N$ | Success Rate (%) | Attacks (%) | $N$ | Success Rate (%) | Attacks (%) |
| 1 | $55.94 \pm 0.24$ | $5.13 \pm 0.86$ | 1 | $87.63 \pm 0.10$ | $33.75 \pm 0.07$ |
| 5 | $56.19 \pm 0.11$ | $23.76 \pm 3.56$ | 5 | $87.92 \pm 0.13$ | $35.39 \pm 0.08$ |
| 10 | $56.43 \pm 0.14$ | $50.50 \pm 4.17$ | 10 | $88.25 \pm 0.13$ | $36.54 \pm 0.10$ |
| 20 | $56.59 \pm 0.00$ | $84.79 \pm 0.93$ | 20 | $88.62 \pm 0.18$ | $39.86 \pm 0.30$ |
| | | | 50 | $89.60 \pm 0.79$ | $51.89 \pm 0.48$ |
| | | | 100 | $92.26 \pm 0.09$ | $71.32 \pm 0.36$ |
| | | | 150 | $94.53 \pm 0.10$ | $91.67 \pm 0.73$ |
| PEA | 56.59 | 100 | PEA | 94.90 | 100 |

component is shared across defenses. Our analysis focuses on the diversity of the space of known attacks designed by the community to be used for a majority of adversarial evaluations.

***Model* vs. *System*.** In commercial applications, ML models often empower an end-to-end system. Such systems can include multiple pre-processing and post-processing steps, which makes analyzing the security of the system a complex task. As most prior work on developing adversarial attacks and defenses focuses on the robustness of the model rather than the robustness of the system, our paper makes the same assumptions. We analyze the diversity and efficiency of model-focused evaluations.

***Input Modality*.** All our evaluations use image classifiers because images are the most common input type used when studying adversarial attacks. However, adversarial attacks against other input modalities have been developed, *e.g.,* text, structured data, audio, *etc.*. Despite a lack of evaluations on these input modalities, we believe our findings will extend to these other modalities, given the similarities in attack methodology.

## 6 Message to the Community

Our paper introduces a new perspective on adversarial evaluation—one that goes beyond simply comparing attack success rates. We believe this more nuanced approach can help the community better interpret and leverage the vast body of both published and unpublished work on the adversarial robustness of neural networks.

***Message to Attackers*.** In the current landscape, most new attacks are solely evaluated by their success rate or speed relative to the state-of-the-art. We encourage researchers to also consider how diverse their attacks are compared to existing ones. The three notions of diversity introduced in this paper provide a useful initial benchmark for assessing such differences. An attack that introduces meaningful diversity along one or more of these dimensions offers greater value to the community, as it enables a deeper understanding of the vulnerabilities in defenses. Right now, we are focused on answering the question of "Is the model vulnerable?" rather than "Why is it vulnerable?". Therefore, the goal should be to develop genuinely diverse attacks rather than merely incrementally more effective ones.

***Message to Defenders*.** Convincing the community that a defense is truly robust has always been a significant challenge. Crafting adaptive attacks often demands substantial effort which, if successful, goes unrecognized. On the other hand, exhaustively evaluating a defense against all existing attacks may seem appealing, and safe, but is often infeasible due to computational constraints and, as we show, redundant. Evaluations should be selective about which attacks to include, prioritizing those that offer genuinely new insights. The notion of diversity introduced in our paper can serve as a valuable guide for diversifying evaluations and communicating results. Defenses can also expand evaluations to include other metrics, such

as the increase in attacker effort or increased attacker visibility. Adversarial robustness is only one aspect of model security.

## 7 Conclusion

Despite more than a decade of research, it is unclear if model security has measurably improved with respect to adversarial evasion. The burden on defenders to ensure the robustness of their defense in all possible situations makes adversarial evaluations highly challenging. Defenders have resorted to running attack ensembles when reporting the empirical robustness of an ML model and defense. However, a lack of formal threat models and an inability to report attack coverage makes it difficult for evaluators to judge the effectiveness of a defense. To avoid publicizing flawed defenses, evaluators are cautious and often request additional empirical robustness evaluation, despite often lacking any technical reason as to why or what specific defensive aspect should be further tested. In this paper, we show that such requests are redundant. Short of adaptive attacks, gradient-based and gradient-free attacks report only slight differences in evaluation precision. When simplifying AutoAttack to a two-attack ensemble, evaluation precision differs by 0.75% on average across 10 distinct defenses. Often, the goal of an evaluation is to answer the question: "Is my model secure?". A 0.75% difference isn't going to affect the overall conclusion. We recognize the importance of comprehensive, precise evaluations of proposed adversarial defenses, but additional requests for attack evaluations should be tactical, not exhaustive.

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
