# OpenReview forum: "Beating a Dead Horse: On the Redundancies of Adversarial Evaluation of Classifiers"
_TMLR — Rejected by TMLR_

### Review · Reviewer_UMmr · 2025-09-24

**Summary Of Contributions:**

This paper explores the diversity of adversarial attacks on adversarial defences. Its main claim is that two attacks are sufficient for evaluating defences. The experiments demonstrate that an attack ensemble of one gradient-based and one gradient-free attack achieves almost the same attack success rate as the four-attack ensemble AutoAttack for current SOTA defences.

**Additional Comments:**

Overall, I am very sceptical of this paper, particularly regarding its story and message. However, I believe that there is great work at its core. The finding on AutoAttack is very interesting, and researching attack diversity is valuable. I recommend substantially reframing the paper and, perhaps, separating positions and findings.

**Audience:**

Yes

**Audience Explanation:**

Evaluating adversarial defences is of interest to people working on adversarial robustness, and the paper's topic is also of interest to the broader AI security community.

**Broader Impact Concerns:**

A core position and motivation of this paper is that "adaptive evaluations" and more exhaustive evaluations using PEA are generally too costly to perform. This is a questionable stance from my perspective. The past has shown that defences are easily broken all too often and all too fast, and need to be evaluated more carefully. From this viewpoint, new adversarial defences simply need to be evaluated more carefully. Perhaps it just requires substantial computational effort to establish that a defence is safe.

From a theory of science standpoint, scientists should not try to prove that their defence is safe. They should try to break it themselves, in the search for a truly safe defence, not with the goal of producing a publication. Since this is an idealised standpoint, the field needs to establish good practices that are expected from authors to push good scientific practice. Requiring "adaptive evaluations" is a push in this direction.

Overall, establishing less rigorous evaluation practices would hurt AI safety overall and is therefore a dangerous direction to pursue. This does not mean that the current standard is perfect and, indeed, two attacks in AutoAttack may be superfluous. However, this does not imply that evaluations should only use two attacks from now on in general, but also, in particular, in light of the results on PEA in the paper.

**Claims And Evidence:**

No

**Claims Explanation:**

**Main issue**
1. While the paper convincingly shows that two attacks are as good as the four-attack AutoAttack, the experimental results on PAE show that there is little attack redundancy on PAE. More precisely, 92% of attacks are required to achieve a comparable success rate as full PAE when attacking the k-winner-takes-all defence (compare to Table 5 and the last paragraph of Section 4.2.1). This shows that it is actually *not* sufficient to execute only two attacks to evaluate defences.

**Further evaluation issues**
 2. The selection of defences in the paper is insufficient for showing the far-reaching claims made in the paper. Evaluating only the currently best-performing attacks may be biased, as these may be highly similar.
 3. Comparing attack diversity based on two high-level semantic class categories in Section 3.3 is not well motivated. Additionally, the observation on airplane misclassification actually shows that there can be meaningful diversity among attacks.
 4. In Section 4.3, the cosine similarity should not be used to compare attacks when the attacks themselves are based on L_p distances, as described in Section 2. Furthermore, cosine similarity ignores perturbation magnitude, which masks likely differences between max-loss and min-norm attacks.

**Requested Changes:**

Besides the issues described above, I have the following requests:

 - The paper title is unnecessarily violent. There are similar idioms that are less graphic. What about "Filling a Bottomless Pit"?
 - The related work should discuss certified defences as well.
 - Please report numbers (equal-hardware costs, runtimes, etc.) to define better what you mean by "computationally expensive".
 - It was frequently unclear to me which defences the paragraphs and figures in the experiments referred to exactly. For example, which defence is evaluated in Figures 3 and 4?
 - The first paragraph of Section 4 says that Section 3 compared $\ell_2$ distances. However, Section 3 only compared cosine distances.


Recommendations (would strengthen the paper):
 - The abstract should be one paragraph.
 - This sentence in the abstract misled me greatly: "There is safety in numbers, and what if additional attacks reveal a lack of diversity in the attack scenarios explored by the original evaluation?"
 - The "names"/citation keys in Table 2 (e.g. "Wong2020Fast") do not add anything to the proper citations following them ("Wong et al., 2019).
 - This paper might profit from being formatted more succinctly.

---

### Review · Reviewer_v65Z · 2025-10-24

**Summary Of Contributions:**

This paper analyzes adversarial attack diversity through 3 outcome metrics (success rate, misclassification, perturbation similarity) and proposes that a two-attack ensemble (one gradient-based, one gradient-free) achieves comparable evaluation precision to AutoAttack with 0.75% average difference,.

Strengths
- The paper defines diversity through measurable outcomes rather than attack components, providing clearer practical evaluation criteria.
- Two-attack ensemble and Algorithm 1 offer actionable computational savings with clear numeric evidence across multiple defenses.
- The paper extends attack space to 164 configurations and evaluates across three complementary metrics within a unified framework.

Weaknesses
- The paper has insufficient generalization evidence. The claims based on comprehensive analysis of only 2 defenses cannot support broad recommendations.
- Only CIFAR-10 for misclassification analysis, single architecture (ResNet-18), and 164 attacks from one framework limit generalizability to other datasets, models, and attack methods.

**Additional Comments:**

1. Could you provide a statistical test to support the reported 0.75–0.79% difference between the two-attack ensemble and AutoAttack?
2. Figure 2 shows that about 47% of gradient-based and 65% of gradient-free attacks achieve over 90% of PEA’s success rate. How can we be confident that the selected attacks are universally
3. In Section 3.3, is the analysis of misclassification diversity possible only for CIFAR-10, where clear semantic categories (animals vs. vehicles)
4. On what basis can the community assume that the two representative defenses sufficiently cover all defense mechanisms?
5. Figure 5 shows that adding a single gradient-free attack to 144 gradient-based attacks dramatically increases the success rate on the k-winners-take-all defense (from ~56% to ~88%). While this demonstrates the value of attack-type diversity, how does it concretely support the paper’s claim of “redundancy” rather than the need for diverse evaluation?

**Audience:**

No

**Audience Explanation:**

- Narrow experimental scope limits applicability, most TMLR readers work on diverse domains/architectures.

- The evdience (Table 5 92% attacks needed for gradient-obfuscated defense) undermines the central "redundancy" message. This makes  confusion to readers.

- Because the study’s results are inherently shaped by its chosen classification framework (gradient-based vs. gradient-free attacks), the paper can be interpreted less as a demonstration of evaluation redundancy and more as an examination of the limitations of that classification framework itself.

**Claims And Evidence:**

Yes

**Claims Explanation:**

- The paper does not cover significance tests or confidence intervals for the central 0.75% "negligible" claim.
- The authors claim based on comprehensive analysis of only 2 defenses.
- The paper tend to write over-generalized. CIFAR-10 only, one architecture, one framework, excludes adaptive/targeted attacks.

**Requested Changes:**

- The authors need to clarify the discussion on how the notion of “redundancy” aligns or conflicts with the community’s emphasis on adaptive and diverse evaluation practices.

- Could the authors provide a clearer explanation of the potential risks of oversimplifying evaluations to two attacks, especially regarding rare or defense-specific vulnerabilities?

---

> ### Author Response · Authors · 2025-12-10
> **Initial Response (1/3)**
>
> Thank you for your insightful comments. We address your queries/concerns below.
>
> ***“The claims based on comprehensive analysis of only 2 defenses cannot support broad recommendations.”***
>
> Our primary claim is that exhaustive evaluations over large non-adaptive attack sets are inherently redundant, because attacks operating within the same search space and using similar search algorithms tend to converge on similar solutions. The purpose of Section 3 is to rigorously test this hypothesis through in-depth examination of the internal diversity of two widely studied attack families (gradient-based and gradient-free) under conditions that intentionally favor each family (i.e., a defense with well-defined gradients and a defense with ill-defined ones, respectively). This allows us to isolate and analyze the intrinsic diversity of the attacks themselves, independent of a particular defense configuration.
>
> Across three definitions of diversity, we consistently observe that attacks that search the input space using similar signals (e.g., gradients) produce highly aligned adversarial outcomes. Within each family, the top-performing attacks converge on nearly identical solutions (i.e., adversarial perturbations), and the long tail of attacks provides negligible new insights into a defense’s robustness. Notably, when evaluated on a defense with well-defined gradients, all gradient-based attacks collapse to the same ineffective result. These findings show that even under two independent defenses with fundamentally different gradient properties, the internal redundancy of the attack families persists. In other words, for both defenses, a small subset of the strongest attacks would have reached the same conclusions as running the entire set.
>
> To evaluate generalizability, we then validate this observation on 11 additional defenses spanning diverse architectures, datasets, and defense mechanisms. Here, we use only a two-attack ensemble — one representative gradient-based and one gradient-free attack — selected based on the insights from Section 3. This extremely lightweight ensemble reaches evaluation outcomes that closely match those of AutoAttack, the state-of-the-art adversarial attack ensemble. These results, reported in Table 4, provide further evidence that the redundancy we identified is not specific to the two initial defenses, but reflects a broader phenomenon across defenses with varied properties. Therefore, the only meaningful diversity we observed across all 164 attacks was gradient-based vs. gradient-free.
>
> Finally, while one could theoretically perform a full exhaustive evaluation across all defenses, doing so would not provide any additional completeness guarantees given the empirical nature of the practice. Our analyses already span defenses with distinct gradient characteristics, threat-model assumptions, model architectures, and datasets, and all yield consistent conclusions. Given this breadth, running comprehensive evaluations on many more defenses would amount to performing redundant experiments without revealing new behaviors in the attack families.
>
> For these reasons, we believe that the claims derived from our focused but comprehensive and principled analyses are sufficiently supported and generalizable.
>
> ***“Only CIFAR-10 for misclassification analysis, single architecture (ResNet-18), and 164 attacks from one framework limit generalizability to other datasets, models, and attack methods.”***
>
> In Section 3, we use only one dataset and one neural network. This section is only used to build evidence for our main hypothesis that gradient-based attacks lack diversity, and so do gradient-free attacks. If our hypothesis is not supported in one scenario, we have no reason to believe it will behave differently in other scenarios. In Table 4, we expand our analysis of the top-10 defenses from the RobustBench leaderboard + one ImageNet defense. Altogether, this table covers variations in neural network architectures, datasets, defense strategies, and vulnerabilities. These results indicate that our learnings from Section 3 and the conclusions they yielded are agnostic to the aforementioned degrees of variation.
>
> Regarding attack selection, Sheatsley et.al. highlighted that all standard attacks can be represented as a permutation of certain “core components”. They derived these core components using a few popular attacks from the literature, and generated ~150 brand new l-infinity gradient-based attacks by permuting these components. We further extended their framework by adding components to support gradient-free attacks. Therefore, the 164 attacks from this framework cover all notable variations in standard, defense-agnostic attacks (e.g., PGD, CW attack, Square attack, etc.). Our paper limits our findings and recommendations to evaluations performed with defense-agnostic attacks. Adaptive attacks require manual development and thus cannot be automated. We’ve noted this as a limitation of our work.

---

> ### Author Response · Authors · 2025-12-10
> **Initial Response (2/3)**
>
> ***"The evdience (Table 5, 92% attacks needed for gradient-obfuscated defense) undermines the central "redundancy" message. This makes confusion to readers."***
>
> The full PEA attack provides the most comprehensive estimate of robustness since it exhaustively executes all available attacks to generate adversarial samples. However, its brute-force nature makes it computationally expensive and impractical in resource-constrained settings. We argue that absolute precision in robustness estimation is not always necessary, especially when similar conclusions can be reached at a fraction of the cost.
>
> Hypothetically, suppose PEA has a 90% success rate against a brand-new road-sign classifier for a self-driving car, but our 2-attack ensemble from Table 4 has only an 88% success rate. In either case, the general conclusion is that the classifier is unusable for deployment. The only difference is that we used significantly more resources in the former than in the latter to arrive at that conclusion. Linking back to Table 5, we can see that only 33% of attacks are needed to identify that the defense is not robust.
>
> A natural concern is whether a smaller ensemble could lead to divergent conclusions. One of our paper’s key objectives is precisely to minimize such divergence by constructing diverse lightweight ensembles. Our 2-attack ensemble in Table 4 exemplifies this: across the top-10 defenses on the RobustBench leaderboard, it achieves attack success rates within 2% of those obtained by the widely adopted AutoAttack ensemble. Although a direct comparison with PEA is computationally infeasible, these results demonstrate that carefully selected, diverse ensembles can serve as efficient and reliable surrogates for large-scale evaluations.
>
> ***“Because the study’s results are inherently shaped by its chosen classification framework (gradient-based vs. gradient-free attacks), the paper can be interpreted less as a demonstration of evaluation redundancy and more as an examination of the limitations of that classification framework itself.”***
>
> The distinction “gradient-based vs. gradient-free” attacks is not something new that we have put forward in this paper. This distinction has emerged over several years of adversarial attack research [[1]](https://arxiv.org/pdf/1902.06705) [[2]](https://arxiv.org/pdf/1810.00069) because attacks have predominantly been developed as search algorithms, and there are two main classes of search algorithms available to us — gradient-based and gradient-free. Collectively, these two classes cover ALL notable attacks in the literature. It is also commonly referred to as white-box vs. black-box attacks.
>
> We use this distinction only to highlight what meaningful attack diversity looks like. While there are several flavors of gradient-based (and gradient-free) attacks, each claiming to be “better” than the others, we show that they appear similar under three definitions of diversity. We also show that the only meaningful diversity occurs BETWEEN a gradient-based and a gradient-free attack as they differ in search methodology.
>
> ***“The authors need to clarify the discussion on how the notion of “redundancy” aligns or conflicts with the community’s emphasis on adaptive and diverse evaluation practices.”***
>
> In the paper, we cover three definitions of diversity: success diversity, prediction diversity, and perturbation diversity. Each definition contributes uniquely towards our understanding of the defense. For example, success diversity allows us to ascertain the worst-case robustness of a defense. Prediction diversity helps us understand the (often spurious) inter-class correlations that our model has learnt. Perturbation diversity provides insights into the types of visual artifacts an attack exploits (e.g., using a blue sky to confuse a bird as a plane), thereby enhancing our understanding of the model’s decision-making process.
>
> The need for adaptive attacks arose from the need to attain greater attack success diversity. By accounting for a defense's inner workings, adaptive attacks can succeed where defense-agnostic attacks fail. Given their defense-specific, non-standard nature, adaptive attacks can not be represented by standardized attack decomposition frameworks like the one we used in this paper. As such, our finding that attacks lack diversity with respect to the aforementioned three definitions applies only to defense-agnostic attacks.
>
> The core message of this paper is that, when it comes to defense-agnostic attacks, one gradient-based and one gradient-free attack exhibit sufficient diversity across all three definitions. If the goal is to seek more success diversity, in order to uncover the defense’s worst-case robustness, manually designed defense-specific/adaptive attacks will need to be included in the attack ensemble as opposed to more defense-agnostic ones.

---

> ### Author Response · Authors · 2025-12-10
> **Initial Response (3/3)**
>
> ***“Could the authors provide a clearer explanation of the potential risks of oversimplifying evaluations to two attacks, especially regarding rare or defense-specific vulnerabilities?”***
>
> Our findings do indeed simplify adversarial evaluations, but only when using standard, defense-agnostic attacks (e.g., PGD, CW attack, Square attack, etc.) We show that one gradient-based and one gradient-free attack is sufficient to make a general determination of the robustness of a defense, which is a valid goal for several use cases as we argue in the paper.
>
> Thus, we would propose a more efficient evaluation pipeline to better allocate evaluation resources. First, evaluate using the two attack ensembles described in the paper. If the required robustness is poor, then the evaluation is concluded as the defense has failed. If it is robust, then create defense-aware attacks rather than performing more standard evaluations. This recommendation is both for defense designers and evaluators. Rare or defense-specific vulnerabilities are not revealed via standard attacks, but via defense-aware adaptations.
>
> ***“Figure 5 shows that adding a single gradient-free attack to 144 gradient-based attacks dramatically increases the success rate on the k-winners-take-all defense (from ~56% to ~88%). While this demonstrates the value of attack-type diversity, how does it concretely support the paper’s claim of “redundancy” rather than the need for diverse evaluation?”***
>
> We would like to point out that the reviewer is misinterpreting the figure respectfully. In fact, our message is the same as the reviewer's: There is meaningful diversity BETWEEN a gradient-based and a gradient-free attack. We only argue that there is NO meaningful diversity WITHIN gradient-based and gradient-free attacks.

---

> ### Comment · Reviewer_v65Z · 2025-12-18
>
> I thank the authors for addressing my comments. I have additional comments below:
>
> 1. While my concerns regarding the first weakness have been addressed, I find the claim in the authors' response to the second weakness logically difficult to accept. (*"If our hypothesis is not supported in one scenario, we have no reason to believe it will behave differently in other scenarios"*). Excluding this particular point, I can understand the rest of the response to the second weakness, and my concerns have been resolved.
> 2. Can the authors clarify in the manuscript that the actual scope of applicability is limited to defense-agnostic attacks only, as mentioned in your response?
> 3. If the authors revise the manuscript, could you please indicate which contents have been modified (e.g., using highlighting)?

---

> > ### Author Response · Authors · 2026-01-29
> > **Follow-up**
> >
> > Thank you for sharing your concerns and action items for the paper! Please see below for our follow-up comments:
> >
> > **Follow-up #1: "While my concerns ..."**
> >
> > *“If our hypothesis is not supported in one scenario, we have no reason to believe it will behave differently in other scenarios.”* - here we are referring to the hypotheses being validated in sections 3.2-3.4, i.e.,
> > - Sec 3.2: Attacks don’t have meaningful diversity in terms of inputs they succeed on
> > - Sec 3.3: Attacks don’t have meaningful diversity in terms of misclassifications they generate
> > - Sec 3.4: Attacks don’t have meaningful diversity in terms of the perturbations they generate
> >
> > Returning to our comment, we posit that if we HAD seen meaningful diversity among attacks based on one of the above three criteria, this would have prohibited us from arriving at our conclusions. But we did NOT witness any such diversity.
> >
> > Moving on to Sec 4, our main focus becomes the attack success rate (i.e., the hypothesis examined in Sec 3.2), as that’s what the community is most interested in. Here, we show on multiple architectures and datasets that a two-attack ensemble is sufficient (Table 4), reinforcing our takeaways from Sec 3.2.
> >
> > Regarding attack selection, to our knowledge, our study is the largest-scale study in terms of the number of attacks examined and executed. To generate such a large set of attacks, we:
> > - Used Sheatsley et. al.’s framework to create a space of attacks based on the fundamental components of the most prominent attacks in literature,
> > - Exhaustively explored the entire space by permuting the components to generate a large set of attacks.
> > - Finally, to ensure that our attack space is representative of (all defense-agnostic) attacks in literature, we also extend their framework to represent gradient-free attacks.
> >
> > No other frameworks (like Sheatsley et. al.’s) could allow us to do this. As such, we believe our attack set accurately represents the space of defense-agnostic attacks.
> >
> > **Follow-up #2: Can the authors ...**
> >
> > We will make sure to explicitly state this in relevant sections of the paper.
> >
> > **Follow-up #3: If the authors ...**
> >
> > We are working diligently to address all issues highlighted in the reviews. Rest assured, we will address all concerns as promised in our response and highlight the changes for easy review.

---

### Review · Reviewer_tQUy · 2025-12-08

**Summary Of Contributions:**

The paper studies:

1. Effect of ensemble of adversarial attacks on attack performance, misclassification outcomes, and input perturbations.

2. Two-attack ensemble consisting of a single well-performing attack from 'each' class performs better than AutoAttack (an ensemble of 4 attacks) performs better only by a smaller margin and does not offer more insight than their method.

**Audience:**

Yes

**Audience Explanation:**

Adversarial attacks are definitely of interest to the community - however I am not sure who the paper is addressed to or what subset of machine learning audience would find the findings interesting from a research perspective. It would be great if the authors can qualify it a bit more.

**Claims And Evidence:**

No

**Claims Explanation:**

The paper focuses on claiming that the primary focus of evaluating ML systems for adversarial robustness is to answer the question "Is the ML system safe?" which is a binary-valued question, and the argument is that with perhaps a much simpler pipeline (of two attacks), one can answer the binary-valued question.

Philosophically, I am skeptical about the premise of the paper in the sense that I don't think any real-world engineering (even non-ML) system is 100% safe, which is why there are different conditions in which a black-box system is rated accordingly. Consider the perhaps more stable system of medicinal drugs, where efficacy is evaluated in terms of percentages (e.g., 80% versus 99.9%). It does matter "how much" a drug is effective. Similarly, one needs to know how accurately how robust a system is. Of course the example that the authors gave of 43% v/s 41% does make sense but the argument relies on a hasty generalization and cherry picking. Its a single example, there can be other examples (e.g. 80% v/s 99.9% robust) where the distinction is clear. Providing evidence for the claims the author makes is, I think, very difficult since one has to test all possible attacks, where the authors are constrained by being able to test only a few state-of-the-art techniques (even if requested) on a couple of datasets for a single task (single-label classification) for a single model and 10 defenses. Therefore, I think it might make sense to reduce the claims accordingly and not make statements around the binary nature of the underlying problem.

Further, as I detail in the changes requested paragraph, I find the paper significantly unclear in its presentation and clarity.

Additionally, the results lack statistical significance - I would assume that more attacks in the ensemble would make it more robust in terms of variance or confidence intervals.

I appreciate the authors for the message put to the attackers and community.

**Requested Changes:**

I would encourage the authors to substantially improve the writing and presentation of the paper.

Section 2.2 and 2.3 might be better in a related work section perhaps?


There are many statements which are provided without evidence (and I think might be untrue).

1. Contributions are not clearly written.
     a. "with 2 defenses with fundamentally different robustness properties" -> Different how in their robustness? what defenses?
     b. "We create a two-attack ensemble consisting of a single well-performing attack from each class and compare
its robustness measurement to AutoAttack across 10 distinct defenses." -> each class? class of the dataset or class with respect to gradient-based and gradient-free attacks?

 2. $L_{CE}$ is not defined before it is first used. I am assuming it is cross-entropy loss.

3. "In both cases, the previously mentioned imperceptibility condition" Its better to reference by an equation number

4. "Most Adversarial ML research focuses on the robustness of image classifiers." -> I would imagine most adversarial ML in the recent (last five years) has been focussed on other modalities including, image generation, text, video, audio etc.

5.  "error rate when under attack is the same: the model is unreliable." -> the model is unreliable by a margin of x% where x% can be possibly large.

It might be useful to add some background on how ensembling of attacks is done mathematically.

For each of the diversity metrics, it might be useful to expand on what it means mathematically and how one exactly partition adversarial methods wrt them.


Further I think the paper can be reorganized to improve the readability substantially more - I think the paper has a nice underlying story and results and can benefit a lot from better exposition of the results.

What is the paper trying to solve -> How -> What are the results. It is right now a lot more convulated and Section 2 and 3 can be improved  & ofcourse the presentation of the plots etc. can be improved a lot.

---

> ### Author Response · Authors · 2025-12-21
> **Initial Response (1/2)**
>
> We thank the reviewer for their valuable feedback on our paper. We address some of the concerns below.
>
> ***“the example that the authors gave of 43% v/s 41% does make sense but the argument relies on a hasty generalization and cherry picking.”***
>
> To contextualize this statement, we would like to bring the reviewer’s attention to Table 2 in the [AutoAttack (AA) paper](https://arxiv.org/pdf/2003.01690). The difference between AA (green column) and the best individual attack (boldfaced) is always within ~2%. Therefore, running AA (4 attacks) as opposed to a single attack does not fundamentally change our understanding of the ~60 defenses that the authors of AA report results for. We will revise the paper to include this argument in replacement of the anecdotal “43% vs. 41%” argument.
>
> Furthermore, as can also be seen from Table 2 of AA paper, the best individual attack is either APGD_DLR (gradient-based) attack or Square (gradient-free) attack in the vast majority of the cases. This two attack ensemble is similar to the two attack ensemble we use Section 4 to highlight the redundancies in prominent attack ensembles. For example, in Table 4 we show that this two attack ensemble comes within 2% of the AAs success rate across top-10 defense from the RobustBench leaderboard. This is consistent with the findings of the AA paper and provides strong evidence for the generalizability of our arguments.
>
> ***“Section 2.2 and 2.3 might be better in a related work section perhaps?”***
>
> We call section 2 as “Background” because it lays the foundations of the problem space we explore in the paper. Related works are mentioned to provide reference for interested readers. It is not positioned as a section to discuss prior works that conduct a similar analysis as ours (none exist as far as we know). Even so, we can rename Section 2 to “Related Works” if that lessens the confusion.
>
> ***“Contributions are not clearly written”***
>
> ***a. "with 2 defenses with fundamentally different robustness properties" -> Different how in their robustness? what defenses?***
>
> The two defenses used in Section 3 differ in which class of attacks they are vulnerable to. First is Fast-AT, which has well-defined gradients and as a result is NOT vulnerable to gradient-free attacks. Second is k-winners-take-all, which has ill-defined gradients and as a result is vulnerable to gradient-free attacks. We can cover these specifics in the intro, we originally skipped them for brevity as they are covered in Section 3.1 (Evaluated Defenses).
>
> ***b. "We create a two-attack ensemble consisting of a single well-performing attack from each class and compare its robustness measurement to AutoAttack across 10 distinct defenses." -> each class? class of the dataset or class with respect to gradient-based and gradient-free attacks?”***
>
> This statement refers to “attack class”, i.e., gradient-based or gradient-free attacks, defined in the preceding paragraph. We will update the text to avoid confusion.
>
> ***“"In both cases, the previously mentioned imperceptibility condition" Its better to reference by an equation number”***
>
> The imperceptibility condition is represented by the norm-constraint on delta within the attack optimization objective (Equations 1 and 2). We will update the text to avoid confusion as
>
> ***“"Most Adversarial ML research focuses on the robustness of image classifiers." -> I would imagine most adversarial ML in the recent (last five years) has been focussed on other modalities including, image generation, text, video, audio etc.”***
>
> Early Adversarial ML research focused on image space attacks as inputs were continuous and differentiable, so attack development was much simpler. While adversarial ML research has been influenced by recent technological developments, attacks in newer domains often involve making adaptations to an image-based attack method. Furthermore, attacks remain scattered across different open source repositories; there is not a well-known collection of attacks in newer domains that offers the same comprehensive coverage as we see in the image domain. The Adversarial Robustness Toolbox and the work by Sheatsley et. al. both provide comprehensive coverage for image based adversarial attack. For all these reasons, we felt it best to present our findings in the image domain.
>
> ***“It might be useful to add some background on how ensembling of attacks is done mathematically.”***
>
> As far as we are aware, there are no mathematically grounded methods for constructing attack ensembles. For example, AutoAttack uses a heuristic notion of attack diversity (white/black-box, loss function usage, and attack objective formulation) to build the ensemble. PEA ensemble is derived by permuting “building blocks” of different attacks.

---

> ### Author Response · Authors · 2025-12-21
> **Initial Response (2/2)**
>
> ***“For each of the diversity metrics, it might be useful to expand on what it means mathematically and how one exactly partition adversarial methods wrt them.”***
>
> We will include mathematical formulas for metrics used in Section 3. Can the reviewer please clarify what they mean by the second half of this comment?
>
> ***Regarding the structure of the paper***
>
> The paper is currently organized as follows:
> - Section 1: Pose Unanswered Questions (What?)
> - Section 2: Background/Pre-requisite knowledge sharing  (What?)
> - Section 3: Our attempt at answering these Questions (How?)
> - Section 4: Testing the generalizability of our answers from the previous section on two prominent attack ensembles (Results)
>
> This aligns with the structure that the reviewer has proposed. Can you please elaborate on what needs to be changed?

---

> > ### Comment · Reviewer_tQUy · 2026-01-20
> >
> > The authors have addressed my concerns. What would be useful is having this organization clear upfront at the end of section 1. I think the writing needs to be improved as the other reviewers have pointed out.
> > Can you address the concerns of reviewer v65Z and update the manuscript with the blue highlighted changes.

---

> > > ### Author Response · Authors · 2026-01-29
> > > **Follow-up**
> > >
> > > Thank you for following up! We are working diligently to address all issues highlighted in the reviews. Rest assured, we will address all concerns as promised in our response and highlight the changes for easy review.
> > >
> > > With regards to paper organization, as we mentioned in our previous comment, to the best of our understanding the current paper organization is already following what the reviewer has proposed. It would be great if you could elaborate on what organizational changes you would like to see so we can adequately address your concerns.
> > >
> > > Please also see our response to reviewer v65Z as we attempt to address their remaining concerns.

---

### Decision · Action_Editor_Tkk1 · 2026-01-31

**Recommendation:** Reject

**Audience:**

Yes

**Audience Explanation:**

The submission focused on an important problem in the community, with diverse metrics for evaluation, which can bring some new insights the community of adversarial attack.

**Claims And Evidence:**

No

**Claims Explanation:**

Two reviewers expressed major concerns about the evaluation scope and the issue of generalization, noting that only two attacks were tested on a single toy dataset. As a result, they feel that the claims made in the paper are not sufficiently substantiated. Although the authors presented a rebuttal, the reviewers maintain that the evidence provided is inadequate to support the paper's contribution.

In particular, reviewer UMmr stated that the paper’s central claims remain insufficiently supported by the current evidence, and emphasized that the limited evaluation scope makes it hard to justify the paper’s broader conclusions.

Reviewer v65Z noted that key issues remained after the response, including limited validation scope and insufficient support for broad generalization claims. The reviewer also asked that the manuscript explicitly scope conclusions to defense-agnostic attacks and strengthen “small difference” claims with statistical testing or uncertainty estimates.

**Resubmission Of Major Revision:**

The authors may consider submitting a major revision at a later time.